# The Endless World of Carotenoids—Structural, Chemical and Biological Aspects of Some Rare Carotenoids

**DOI:** 10.3390/ijms24129885

**Published:** 2023-06-08

**Authors:** Nikolay E. Polyakov, A. Ligia Focsan, Yunlong Gao, Lowell D. Kispert

**Affiliations:** 1Institute of Chemical Kinetics & Combustion, Institutskaya Str. 3, 630090 Novosibirsk, Russia; polyakov@kinetics.nsc.ru; 2Department of Chemistry, Valdosta State University, Valdosta, GA 31698, USA; alfocsan@valdosta.edu; 3College of Sciences, Nanjing Agricultural University, Nanjing 210095, China; yunlong@njau.edu.cn; 4Department of Chemistry and Biochemistry, The University of Alabama, Tuscaloosa, AL 35487, USA

**Keywords:** rare carotenoids, xanthophylls, electron transfer, radical cation, carotenoid complexes, hydrogen bonding, nonpolar environment

## Abstract

Carotenoids are a large and diverse group of compounds that have been shown to have a wide range of potential health benefits. While some carotenoids have been extensively studied, many others have not received as much attention. Studying the physicochemical properties of carotenoids using electron paramagnetic resonance (EPR) and density functional theory (DFT) helped us understand their chemical structure and how they interact with other molecules in different environments. Ultimately, this can provide insights into their potential biological activity and how they might be used to promote health. In particular, some rare carotenoids, such as sioxanthin, siphonaxanthin and crocin, that are described here contain more functional groups than the conventional carotenoids, or have similar groups but with some situated outside of the rings, such as sapronaxanthin, myxol, deinoxanthin and sarcinaxanthin. By careful design or self-assembly, these rare carotenoids can form multiple H-bonds and coordination bonds in host molecules. The stability, oxidation potentials and antioxidant activity of the carotenoids can be improved in host molecules, and the photo-oxidation efficiency of the carotenoids can also be controlled. The photostability of the carotenoids can be increased if the carotenoids are embedded in a nonpolar environment when no bonds are formed. In addition, the application of nanosized supramolecular systems for carotenoid delivery can improve the stability and biological activity of rare carotenoids.

## 1. Introduction

Carotenoids are a class of more than 1200 naturally occurring pigments synthesized by plants, algae and photosynthetic bacteria [1]. Carotenoids are important for human health because they act as antioxidants, helping to protect cells from damage caused by reactive oxygen species (ROS). They have been associated with a reduced risk of chronic diseases such as type 2 diabetes [2], cancer [3,4], heart disease or age-related macular degeneration [5]. Another important role is their provitamin A activity or the capability of some dietary carotenoids to form vitamin A by the action of dioxygenase enzymes [6]. For example, the dioxygenase enzyme β-Carotene 15-15′-oxygenase (BCO1) catalyzes the oxidative cleavage of dietary β-Carotene to retinal (vitamin A aldehyde) [7], which can be further reduced to retinol (vitamin A) or oxidized to retinoic acid (the biologically active form of vitamin A). Carotenoids such as β-carotene and α-carotene β-cryptoxanthin are considered provitamin A carotenoids, but any carotenoid with at least one unchanged β-ionone ring in its structure can have provitamin A activity. In addition to their health benefits, carotenoids in plants are also involved in photosynthesis. They are no longer considered just accessory pigments [8]; they have essential roles in photosynthesis [9], helping to capture light energy and transfer it to chlorophyll molecules and also protecting plants from damage caused by excess light and other environmental stresses [10]. Similarly, in the human body, carotenoids photoprotect against damage by intense light and harmful free radicals, and also maintain the structural and functional integrity of biological membranes. The mechanisms through which carotenoids may exert their health effects are complex, and further studies, including clinical studies, are needed to provide a comprehensive understanding. Carotenoids are indispensable for life. The world of carotenoids is endless. Their number and resourcefulness are immense, and only a tiny fraction of the 1200 carotenoids has been studied to date. Their complexity lies not only in their number, but in their different yet alike structures (given by the different functional groups and similar polyene chains) and their interactions with a multitude of different environments. 

In the past 20 years, we have gained significant understanding about carotenoids’ physicochemical properties and their interaction with other compounds in different environments [11]. We have mostly used advanced electron paramagnetic resonance (EPR) analysis techniques, such as continuous wave or pulsed electron nuclear double resonance (ENDOR), in combination with density functional theory (DFT) molecular orbital calculations, to elucidate the physisorption and electron and proton transfer processes that occur when carotenoids are adsorbed on solid artificial matrices [11]. Similar reactions were predicted in solution from electrochemistry studies performed in the 1990s, or in vivo in more recent studies performed in the 2000s. Additionally, EPR spin-trapping studies have been performed to characterize the inclusion complexes of carotenoids with different delivery systems [11]. In this paper, we review EPR and DFT studies relevant to understanding carotenoid chemistry, and we present the structural, chemical and biological aspects of several rare carotenoids that need to be considered when designing new systems for carotenoid delivery.

## 2. Studies of Electron and Proton Loss of Conventional Carotenoids by DFT and EPR

Carotenoids are prone to oxidation. We have studied the oxidation of carotenoids adsorbed on solid artificial matrices, such as silica alumina or imbedded in the pores of molecular sieves MCM-41, where their radical cation is formed by electron transfer from the carotenoid molecule to the matrix. Some of the most well-known carotenoids that we studied using EPR methods in combination with DFT include lycopene, β-carotene, zeaxanthin, canthaxanthin, lutein, astaxanthin and *cis*-bixin (Figure 1). With normal EPR at the X-band frequency (9 GHz), the carotenoid radical cation exhibits a single unresolved peak with g_iso_ = 2.0027, characteristic of organic π-radicals. In the year 1999, the EPR signal previously not resolved at the X-band frequency was resolved at a higher frequency (330 GHz) [12]. At higher frequencies, from 327 to 670 GHz, the unresolved line resolves into two peaks as a result of the g-anisotropy of g_⊥_ = 2.0023 and g_||_ = 2.0032, characteristic of a cylindrically symmetrical π-radical cation. Determining the g-tensors from high-field spectra is important for learning about molecular structure from its principal values. The difference g_xx_ − g_yy_ decreases with increasing chain length. When g_xx_ − g_yy_ approaches zero, the g-tensor becomes cylindrically symmetrical with g_xx_ = g_yy_ = g_⊥_ and g_zz_ = g_||_. This applies for the all-*trans* carotenoid radical cations and allows differentiation between carotenoid radical cations with cylindrical symmetry and other C-H organic radicals of different symmetry. The lack of temperature dependence of the EPR line widths over the range of 5–80 K at 327 GHz also suggests a rapid rotation of methyl groups even at 5 K, which averages out the proton couplings from three oriented β-protons. This results in isotropic β-proton couplings from rotating methyl groups [12].

Even though the number of hyperfine couplings is greatly reduced when considering this rapid rotation of the methyl groups, carotenoid radicals still contain a large number of anisotropic α-protons which give rise to numerous anisotropic coupling constants. These couplings cannot be determined by normal EPR. Instead, ENDOR techniques can be used to determine the hyperfine couplings of carotenoids adsorbed on silica alumina or in MCM-41. Continuous wave and pulsed ENDOR showed that for irradiated carotenoids on solid matrices, not only radical cations were formed, but also neutral radicals formed by the deprotonation of the radical cation, which is a weak acid. These neutral radicals formed by proton loss from the radical cations contain lots of similar hyperfine couplings to those of the radical cation, but ENDOR techniques helped to distinguish the two different radical species [11]. It is important to note that the presence of all radicals is enhanced by the irradiation of samples and the presence of metals, such as in metal-substituted MCM-41. The deprotonation of the radical cations to form neutral radicals determined by ENDOR on solid surfaces, which was also proven electrochemically in solution, needs to be considered in vivo where the radical cation is formed and is known to have a role in photoprotection mechanisms. We have also hypothesized that neutral radicals could have a role in an additional quenching mechanism to that of the radical cation [13].

DFT calculations were used starting in the early 2000s to determine the hyperfine coupling constants of radical cations and neutral radicals and to simulate spectra which matched the experimental ENDOR spectra and confirmed the identity of the radicals. DFT was also used to predict the most favorable positions for proton loss from the radical cation and establish the relative stability of the neutral radicals formed from the radical cation, as described next. Table 1 indicates the most favorable positions of proton loss from the radical cation for carotenoids listed in Figure 1.

Lycopene is a symmetric linear carotenoid that has ten methyl groups in four distinct positions: C1(1′), C5(5′), C9(9′) and C13(13′) (see Figure 1). The primed positions of this molecule are equivalent by symmetry to the unprimed positions. There is a smooth relationship between the relative energy ΔE(n) of a neutral radical formed by proton loss from the radical cation, and the conjugation or delocalization length, N, over which the unpaired spin density is distributed. The longer the conjugation length, the most stable the radical is. DFT has shown that the most stable neutral radicals for lycopene are formed by proton loss at the C4 or C4′ methylene positions, which extend conjugation. It is thus expected that proton loss occurs more favorably from the C4(4′) positions [14]. For β-carotene, which has two symmetric cyclohexene rings at the ends of the molecule, proton loss occurs also at the C4 methylene position, and symmetrically at C4′, rather than the methyl groups attached at C5(5′), C9(9′) and C13(13′) [15].

Zeaxanthin has the same structure as β-carotene with two additional hydroxyl groups on positions C3 and C3′, respectively. The two hydroxyl groups have no effect on the position of proton loss from a radical cation, so the most favorable neutral radicals form by proton loss at the C4(4′) methylene positions [16]. However, when the C4(4′) methylene positions contain carbonyl groups, for example in the case of canthaxanthin, proton loss occurs from the methyl groups attached at positions C5(5′). Lutein, an isomer of zeaxanthin which differs from it by one shifted double bond at C4′-C5′ (instead of C5′-C6′ in zeaxanthin), makes proton loss more favorable at the C6′ position instead of the C4′ position [15]. When both hydroxyl and carbonyl groups are present, such as in astaxanthin, it is possible for proton loss to occur at the C3 and C3′ positions [17]. Bixin is a C25 carotenoid, with only nine conjugated double bonds and thus a shorter-length carotenoid compared to the C40 structure of the others presented here. It is an asymmetric carotenoid with -COOH at one end and -COOCH_3_ at the opposite end. Its IUPAC name is 6-methyl hydrogen (9′Z)-6,6’-diapocarotene-6,6′-dioate [1]. According to [18], bixin was the first *cis*-carotenoid to be isolated from natural sources and *trans*-bixin is a more stable isomer that the *cis* form. DFT calculations by Hernandez-Marin et al. [19] also show that, in most cases (with the exception of 13-cis auroxanthin), out of 11 carotenoids studied, the *trans* isomers are more stable than their corresponding 9- and 13-*cis* isomers, while the 15-*cis* isomers are the least stable isomers. However, upon oxidation of the neutral molecule to form the radical cation, our DFT calculations [20] show that the radical cation of *cis*-bixin becomes more stable than the *trans* radical cation of bixin. Furthermore, proton loss from *cis*-bixin occurs from the methyl group the C9′ position of the *cis*-bixin radical cation, on the side with the acetate group [20].

Proton loss from the radical cations of carotenoids has enabled us to predict the most acidic protons which are usually at the ends of the carotenoid to extend the conjugation length [14]. We have hypothesized that zeaxanthin and lutein radical cations’ ability to deprotonate in light harvesting complex II (LHC II) and form neutral radicals could be linked to their quenching activity [21].

**Table 1 ijms-24-09885-t001:** The most favorable proton loss positions for selected carotenoids in our studies.

Carotenoid	The Most Favorable Proton Loss from the Radical Cation	Reference
Lycopene	C4(4′)	[14]
β-carotene	C4(4′)	[15]
Zeaxanthin	C4(4′)	[15,16]
Canthaxanthin	C5(5′)	[15]
Lutein	C6′	[15]
Astaxanthin	C3(C3′)	[17]
*cis*-Bixin	C9′	[20]

## 3. Improvement of Carotenoid Features by Nonpolar Environment and Absence of Hydrogen Bonding—Rare Carotenoids Discussion

The structure and type of a carotenoid (polar/nonpolar) and its position/orientation in the type of environment (polar/nonpolar) are also extremely important for the role that the carotenoid plays. The oxygen-containing groups (hydroxyl, carbonyl, carboxyl, acetate) situated at the ends of carotenoids can not only affect the position of proton loss as discussed above, but also play a role in anchoring the carotenoid in a certain environment.

DFT calculations, calorimetric experiments and EPR studies confirmed that carotenoids containing oxygen groups can form hydrogen bonds (H-bonds) on the surfaces of hosts [22]. This paper predicts the behavior of some rare carotenoids (Figure 2) containing oxygen groups.

Compared with conventional carotenoids, the rare carotenoids usually contain more oxygen-containing groups such as hydroxyl, keto, ester and ether groups, or a similar number of groups but situated somewhere else than the ring (see Figure 2). Both deinoxanthin and siphonaxanthin contain hydroxyl groups and a keto group. Deinoxanthin contains two hydroxyl groups and one keto group, while siphonaxanthin contains three hydroxyl groups and one keto group. Sioxanthin contains five hydroxyl groups and two ether groups; crocin contains fourteen hydroxyl groups, two ester groups, two keto groups and six ether groups.

It was confirmed that canthaxanthin can form hydrogen bonds (H-bonds) on surfaces of hosts such as MCM-41 [22]. For example, each canthaxanthin molecule can form two H-bonds on the surface of MCM-41 (see Figure 3). Hydrogen bonds are formed with the surface silanol groups in MCM-41.

If rare carotenoids with oxygen-containing groups like the ones described above are incorporated in host molecules, we hypothesize that multiple H-bonds can be formed (see Figure 4 for the rare carotenoid complexes). For example, deinoxanthin can form three H-bonds, and acts as both a H-bond donor and acceptor in the complex; myxol also forms three H-bonds, but can only act as a H-bond donor.

The carotenoids can be better stabilized when multiple bonds are formed. Our previous study [23] shows that both the HOMO and LUMO energies of carotenoids increase when the carotenoids act as H-bond donors, and the opposite is true if the carotenoids act as H-bond acceptors. Figure 5 shows the two modes for retinol adsorbed on the surface of MCM-41. DFT calculations demonstrated that both the HOMO and LUMO energies of retinol increase by ~0.2 eV compared with the free molecule when retinol acts as a H-bond donor, and the energies decrease by about ~0.1 eV when retinol acts as a H-bond acceptor [23]. This effect should be more pronounced if multiple H-bonds are formed (e.g., myxol acts as a H-bond donor in the 3 H-bonds and crocin acts as a H-bond donor in the 14 H-bonds, see Figure 4).

A study by Méndez-Hernández et al. [24] shows that there is a very strong linear correlation between DFT-calculated HOMO and LUMO energies (HLE) and the redox potentials of 51 polycyclic aromatic hydrocarbons (PAHs). The strong correlation obtained from the HLE and redox potentials of PAHs was independent of whether the solvent model was included in the calculations. This is consistent with our previous study [25], which shows that the oxidation potentials of carotenoids increase with decreasing the HOMO energies of the carotenoids. Carotenoids are more stable if their oxidation potentials are high because the carotenoids are less likely to be oxidized by metal ions, such as Cu^2+^ and Fe^3+^. The antioxidant activity of a carotenoid’s supramolecular complex is also high if the oxidation potential of the carotenoid is high because the scavenging ability of a carotenoid towards the free radicals increases nearly exponentially with increasing carotenoid oxidation potential, and the increase of 0.03–0.05 V causes the scavenging rate constant to increase about 30 times [26,27]. Our previous study also shows that the orientation of a carotenoid in the host molecule affects the antioxidant activity [25]. It is determined that free radicals abstract the most acidic hydrogen in the conjugated system of a carotenoid [27,28]. If access to the most acidic hydrogen is blocked for the free radicals, the antioxidant activity of the carotenoid decreases significantly. For instance, the most acidic hydrogen of deinoxanthin is at C3 (see Figure 4). If the hydrogen is close to the surface of the host molecule after the formation of the H-bonds, access to the hydrogen may be blocked, resulting in low antioxidant activity.

**Figure 4 ijms-24-09885-f004:**
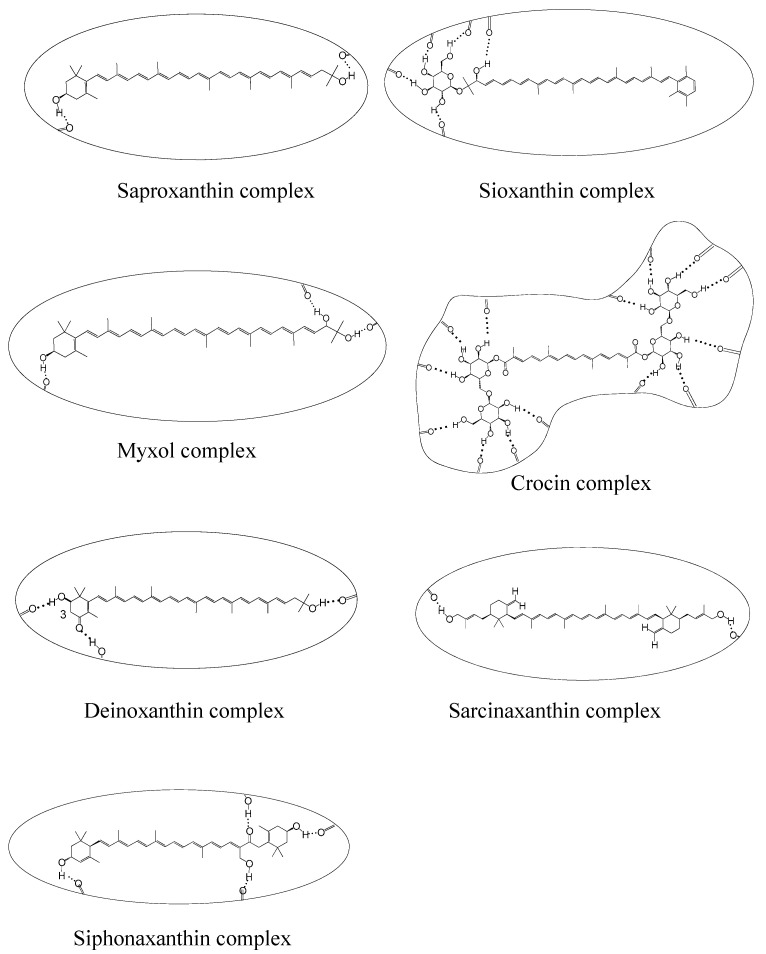
Complexes of the rare carotenoids indicating H bonding.

**Figure 5 ijms-24-09885-f005:**
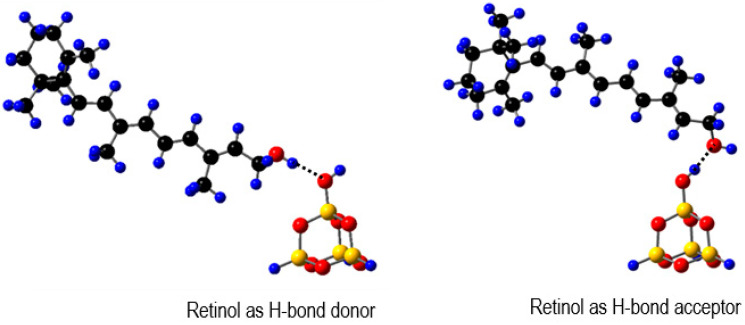
Two types of H-bonds for retinol adsorbed on the surface of MCM-41. H: blue, O: red, C: black and Si: yellow. Adapted from Ref. [23].

Our DFT calculations [23] also show that when a carotenoid acts as a H-bond acceptor, the formation of the H-bond decreases the energy of the LUMO of the carotenoid, which also stabilizes the neutral species more than the radical cation, disfavoring the photo-induced ET from the carotenoid to host molecules. This effect is more pronounced if the oxygen atom of the carotenoid is completely conjugated with the conjugated chain. However, if the carotenoid is a H-bond donor, the formation of the H-bond increases the energy of the LUMO of the carotenoid, which also stabilizes the radical cation more than the neutral species, thus significantly increasing the charge separation efficiency. The photoinduced electron transfer affects the photostability of carotenoids, which is related to the formation of carotenoid radicals [29]. Carotenoid radical cations are formed by photoinduced electron transfer to electron acceptors, and their deprotonation in the presence of proton acceptors creates carotenoid neutral radicals which are highly reactive species that can form other products as well as carotenoid dimmers [30]. It was also concluded in our study [31] that the photostability of a carotenoid is the highest if the environment of the carotenoid is nonpolar and no hydrogen bonds are formed because the electron transfer is intermolecular and thus is slow. There is an approximately exponential decrease in ET rate with increasing distance [32,33,34,35]. Another reason for the lower charge separation efficiency is that the nonpolar environment stabilizes the radical cation of the carotenoid less efficiently than the polar environment, according to our DFT calculations [31]. However, low photostability may be an advantage in some cases, such as in phototherapy, when radical cations or neutral radicals of carotenoids are needed. The rare carotenoids may contain several hydroxyl groups (e.g., crocin contains 14 hydroxyl groups) that can act as H-bond donors in the host molecules, and would be easily photo-oxidized. The rare carotenoids are good candidates for this purpose. The surface of a host can be modified so that a carotenoid can act as only a H-bond donor or as an acceptor. For example, when the surface functional group Ti-OH is replaced by Ti-F by ion exchange, 7-hydroxycoumarin (7-CN) acts as both a H-bond donor and acceptor on the surface of TiO_2_, but acts as only a H-bond donor on F-TiO_2_ (see Figure 6). EPR measurements showed five-fold increases in free radical yields for 7-CN on F-TiO_2_ compared with TiO_2_. DFT calculations for the 7-CN on TiO_2_ and F-TiO_2_ were performed to investigate these phenomena [36]. The calculations show that when 7-CN act as the H-bond donor, the driving force for photo-induced electron transfer from the dyes to TiO_2_ is higher, and the dye’s excited state mixes strongly with the TiO_2_ conduction band states. This is attributed to the shorter distance between the coumarins and the surface of TiO_2_ when coumarins act as the H-bond donors [36].

Different anchoring modes of carotenoids on the surface of a semiconductor affect the bond length, degree of conjugation, energies of the HOMO and LUMO and the mixing of the carotenoids’ excited states with the conduction band states of the semiconductor [37]. For example, three anchoring modes were observed for retinoic acid (RA) on the surface of TiO_2_ (see Figure 7). Mode A: the carboxylic acid group loses a proton and forms two coordination bonds with two surface Ti atoms of TiO_2_. Mode B: the carboxylic acid group loses a proton and forms one coordination bond with one surface Ti atom of TiO_2_ and one H-bond with a Ti-OH group on the surface of TiO_2_. Mode C: the carboxylic acid group does not lose a proton and forms one coordination bond with one surface Ti atom of TiO_2_ and one H-bond due to the interaction between the OH group of the carboxylic acid group and the bridging oxygen atom on the surface of TiO_2_. The degree of conjugation in RA increases from B to A to C. TD-DFT calculations [37] showed that compared with the free carotenoid, the maximum absorption wavelengths (λ_maxs_) of A, B and C shift to the red because the degree of conjugation of RA anchored on TiO_2_ increases, which is confirmed by the different colors of the samples. Mode B involving the surface Ti-OH group has the highest driving force for photo-induced ET. This is attributed to the lowest degree of conjugation of RA for this mode and the highest mixing of the RA’s LUMO with the conduction band states of TiO_2_.

Carotene carotenoids can also form complexes with metal ions through metal–olefin interaction [38]. β-carotene can form a complex with Cu^2+^ on the surface of Cu-MCM-41, as examined by the EPR study [38]. Figure 8 shows the optimized complex from DFT calculations [39]. The EPR study showed that the formation of the complex favors the photoinduced electron transfer from β-carotene to Cu^2+^ and also permits thermal back electron transfer from Cu^+^ to β-carotene radical cation [38]. However, xanthophyll carotenoid canthaxanthin does not form a complex with Cu^2+^ on the surface of Cu-MCM-41. DFT calculations showed canthaxanthin prefers to form H-bonds with the surface silanol groups (−SiOH) because the interaction energy is higher than that between canthaxanthin and Cu^2+^ [22]. It is worth studying the rare carotenoids on the surfaces of semiconductors. For instance, deinoxanthin may form both coordination bonds and H-bonds on the surface of a semiconductor. It is important to know how the formation of those bonds affects the degree of conjugation, energies of the HOMO and LUMO, the mixing of the carotenoids’ excited states with the conduction band states of the semiconductor, and thus the photoinduced charge separation of the carotenoids. There may be steric hindrance for the interactions, depending on the types of host molecules. Choosing appropriate host molecules for semiconductors can remove or mitigate the steric hindrance.

## 4. Improvement of Carotenoid Features by Supramolecular Delivery Systems

In addition to the previously studied carotenoids shown in Figure 1, the rare carotenoids shown in Figure 2 have similar limitations in practical applications. They are scarcely stable and lose most of their functionality after exposure to light, heat, oxygen and acids. Moreover, due to their low solubility in water, they have poor absorption and low bioavailability after oral administration. The bioavailability of carotenoids is affected not only by the type and amount of carotenoid but the molecular linkage, the matrix in which the carotenoid is incorporated, effectors of absorption and bioconversion, the nutrient status of the host, genetic factors, host-related factors and mathematical interactions [6].

Thus, despite their beneficial effects, chemical instability, low bioavailability and high susceptibility to process conditions drive researchers to find appropriate approaches to overcome the above-mentioned obstacles. From our own experience and published data of other authors [40,41,42,43,44,45,46,47], the use of supramolecular delivery systems can help to solve most of the above-mentioned problems of carotenoid applications in the medicine, cosmetics and food industries. In addition, insight from our EPR experiments has allowed us to find some unusual effects of supramolecular delivery systems on the physicochemical and photochemical properties of carotenoids, as discussed below. In this part, we will also briefly describe the biological activities of some rare carotenoids shown in Figure 2, the supramolecular systems used for carotenoids delivery and how their application can help to improve the stability and biological activity of rare carotenoids.

### 4.1. Biological Activities of Rare Carotenoids

Rare carotenoids, including siphonaxanthin, saproxanthin, myxol, sioxanthin and some others, contain oxygen groups and belong to the xanthophyll carotenoids. They are mainly present in some bacteria and marine algae. The most important feature of carotenoids is their antioxidant activity. Carotenoids are the most efficient natural quenchers of singlet oxygen due to their conjugated double bonds system, and this quenching ability increases with increasing the number of conjugated double bonds. Taking into account that singlet oxygen is the main reactive oxygen species (ROS) responsible for photo-oxidative damage to cell membranes and human tissues, carotenoids might have beneficial effects on skin protection [48]. Due to their electron-rich structure, carotenoids can also scavenge other free radicals, including hydroxyl and peroxyl radicals, as well as free radicals of xenobiotics [49,50,51]. The strong antioxidant activity of carotenoids reduces the risk of many types of diseases caused by free radicals, such as cardiovascular disease, cancer and other age-related diseases [52]. Assuming the antioxidant ability of carotenoids as an important factor against oxidative stress, the statistical data indicated a direct correlation between the use of carotenoids in diets and a decreased incidence of cancer types. Rare marine carotenoids show great chemical diversity, resulting in potential novel therapeutic properties, but many of these novel properties are still the subject of future studies.

Siphonaxanthin shows beneficial effects on health, including anticancer activity in the treatment of leukemia [53]. Additionally, it shows anti-angiogenic, antioxidant and anti-inflammatory activities [54,55]. Saproxanthin and myxol are potent antioxidants. They show high antioxidant activity against lipid peroxidation in the rat brain homogenate model and have a neuroprotective effect on L-glutamate-induced toxicity [56,57]. Myxol is also effective in strengthening biological membranes, reducing permeability to oxygen [58]. Deinoxanthin is a unique carotenoid synthesized by *Deinococcus radiodurans*, one of the most radioresistant organisms known for its high resistance to stresses including radiation and oxidants. Deinoxanthin exhibits significantly stronger ROS-scavenging activity in comparison with other carotenoids [59]. Using quantum chemical calculations, it was found that this carotenoid possesses lower triplet excitation energy than other carotenoids, such as β-carotene and zeaxanthin, which provides its strong potential in the energy-transfer-based ROS-scavenging process. In addition, authors show that the H-atom donating potential of deinoxanthin is also crucial for its strong antioxidant activity. On the other hand, an in vitro study demonstrates the novel functional property of deinoxanthin as a potent inducer of apoptosis in human cancer cells. Deinoxanthin treatment caused an increase in ROS in cancer cells, suggesting possible pro-oxidant activity [60]. The authors suggest that deinoxanthin could be potentially useful as a chemopreventive agent. When carotenoids are delivered with ROS-inducing cytotoxic drugs, they can minimize the adverse effects of these drugs on normal cells by acting as antioxidants and minimizing oxidative stress, without interfering with their cytotoxic effects on cancer cells as pro-oxidants which enhance oxidative stress in cancer cells [61]. 

Sioxanthin is a C_40_ carotenoid, glycosylated on one end of the molecule. Glycosylation is an unusual feature among carotenoids and sioxanthin represents a poorly studied group of carotenoids which are polar on one end and non-polar on the other [56,62]. Crocin, in contrast to all other carotenoids, is a water-soluble carotenoid glycosylated on both ends of the molecule [63]. Crocin is effective as an antioxidant and as a learning and memory enhancer, and it shows activity against brain neurodegenerative and Alzheimer’s disorders. Crocin also shows an anti-angiogenesis effect and is likely to be involved in the regulation of molecules in the angiogenesis pathway [64]. However, similar to other highly bioactive carotenoids, it has limited use due to its instability at low pH and during heat and oxidative stress, and its low bioavailability, and is a unique C50 carotenoid isolated from marine bacteria, which shows high antioxidant activity in the singlet oxygen (^1^O_2_) quenching model [65].

Table 2 summarizes the biological roles of rare carotenoids described above. It can be concluded that although antioxidant activity is a common property of most carotenoids, its specific manifestation depends on the localization of the carotenoid in the cell and on its oxidative potential. As for the other properties of carotenoids, they significantly depend on the chemical structure.

### 4.2. Supramolecular Delivery Systems of Carotenoids

The incorporation of carotenoids in nanosized supramolecular carriers can significantly change their physicochemical and biological properties, as well as their therapeutic potential. The nanoencapsulation of carotenoids is an effective strategy to improve their stability towards heat, light, oxygen, metal ions and processing conditions, as well as to increase water solubility [44,45,46,47]. In addition, nanoencapsulation can change the oxidation potentials of carotenoids [42], their absorption spectra [27] and their ability to form J- and H-type self-aggregates [27]. The latter property is especially important for xanthophyll carotenoids [66,67]. Additionally, nanoencapsulation enhances the bioavailability of carotenoids via modulating their release kinetics from the delivery system. The important property of supramolecular carriers is the possibility to modify their surface by attaching new functional groups, providing targeted delivery to the necessary organs or cell receptors [68,69]. The design of functional nanocarriers enables even on-demand drug release in specific microenvironments of various diseases. Nowadays, nanocarriers include lipid nanoparticles, polymers, micelles, inorganic nanoparticles, hybrid nanoparticles and others. Detailed descriptions of various nanocarriers can be found in numerous reviews [44,45,46,47]. Here, we will just briefly describe two delivery systems, polysaccharide arabinogalactan (AG) and saponin glycyrrhizin or glycyrrhizic acid (GA) (see Figure 9), studied earlier by EPR and optical techniques [40,41,42,43].

Glycyrrhizin (GA) is the main active component of licorice root, and has frequently been used in traditional Persian, Chinese and Japanese medicine [70,71]. Recent studies have also shown its antioxidant [72], anticancer [73] and antivirus activities, including SARS-CoV-2 virus [74,75]. In addition, it was demonstrated that due to its amphiphilic nature, GA is able to form micelles in aqueous solutions, as well as water-soluble supramolecular complexes with hydrophobic drugs, including carotenoids [76]. Furthermore, the membrane-modifying ability of GA has been discovered as one of the key mechanisms of its activity [77,78].

The encapsulation of the xanthophyll carotenoids lutein and zeaxanthin into GA micelles protects these carotenoids from oxidation by ROS and metal ions [43]. It was demonstrated that GA forms supramolecular complexes with carotenoids, not only in water solutions with increasing carotenoid solubility by several orders, but also in non-aqueous organic solvents such as methanol, DMSO and acetonitrile [41]. This GA property is important for understanding the GA-assisted transport of carotenoid molecules through lipophilic cell membranes and their membrane protection properties. One of the most important biological properties of carotenoids is their antioxidant activity. Using the EPR spin-trapping technique, a synergetic effect of GA on the scavenging rates of hydroperoxyl radicals by carotenoids has been demonstrated [42]. The strongest increase in scavenging rates (20–30 times) was observed for carotenoids with high oxidation potentials. Since the scavenging rate is strongly potential-dependent, we suggested that GA affects the oxidation potential of the carotenoids. Electrochemical measurement of the oxidation potentials of the carotenoids zeaxanthin and canthaxanthin confirms this hypothesis. The increase in the oxidation potentials by 0.03–0.05 V was observed in the presence of GA [42].

Another characteristic feature of xanthophyll carotenoids is the ability to form J- and H-type aggregates in aqueous solution and in lipid membranes [51,52]. EPR spin-trapping and optical studies have shown that the formation of such aggregates significantly reduces the antioxidant and photoprotection abilities of these carotenoids [17]. The complexation of these carotenoids with GA reduces the aggregation ability of the xanthophyll carotenoids, and simultaneously increases their scavenging ability towards oxygen radicals. Taking into account the wide spectrum of biological activity of xanthophyll carotenoids, including eye and skin protection, GA could be considered a prospective carrier for rare carotenoids.

Arabinogalactan (AG) is a natural branched polysaccharide with a molecular mass near 16 kDa extracted from Siberian larch [79]. AG is a highly water-soluble polymer which produces low-viscosity water solutions. It is biodegradable, biocompatible and contains different functional groups, such as hydroxyl and carboxylic acid, that make it suitable for conjugation with functional groups for targeted delivery. The solubility of carotenoids complexes with AG prepared by the mechanochemical technique [40] were 2–5 mM in aqueous solution, which is several orders of magnitude higher than the solubility of free carotenoids in pure water (~1 nM) [80]. The mechanochemical method allows the preparation of inclusion complexes of drugs with natural or synthetic polymers in a solid state in one technological step without using any organic solvents [79]. It was shown that complexation with AG prevents the H-aggregate formation of xanthophyll carotenoids in the presence of water, similarly to what was detected for GA complexes [17]. Furthermore, the xanthophyll carotenoids lutein, astaxanthin and canthaxanthin have demonstrated an enhancement of photostability (5–10 times) and oxidation stability in AG complexes [40,43]. We assume that the main mechanism of the enhanced photostability and oxidation stability of these carotenoids in the polymer nanoparticles is their isolation from water molecules by incorporation into the hydrophobic polymer environment. As demonstrated by various EPR techniques, the oxidation of carotenoids results in the formation of carotenoid radical cations, which are unstable in aqueous solutions due to fast deprotonation and the formation of unstable neutral radicals [30,81]. The high effectiveness of AG as a delivery system for zeaxanthin and lutein was also confirmed in vivo using wild-type mice as a model [82]. Supplementation with these macular carotenoids can prevent and reduce the risk of age-related macular degeneration and other ocular diseases. Significant increases in lutein and zeaxanthin amounts in serum, liver and RPE/choroid of the mice were detected when they used the carotenoid-AG complex.

An important property of the carotenoid-AG inclusion complex, discovered by the EPR technique, is the stabilization of carotenoid radical cations. This feature has been demonstrated using EPR and ESEEM (electron spin echo envelope modulation) techniques under UV irradiation of the canthaxanthin-AG complex on the surface of titanium dioxide nanoparticles [40]. The carotenoid-AG complex, when irradiated on TiO_2_, shows a significant increase in the EPR signal intensity of carotenoid radical cations compared to that of pure carotenoids. TiO_2_ nanoparticles are widely used in photocatalysis and in artificial solar cells due to their ability to absorb light and to transfer negative or positive charge to a corresponding electron acceptor or donor [83]. The authors of [40] suggested that the significant increase in the yield of radical cations is due to the isolation of the carotenoid radical cation from the surface of TiO_2_ by incorporation into the polysaccharide matrix. It allows more efficient charge separation and reduces the rate of back-electron transfer. Note that such a method of light energy transformation is similar to the mechanism used by plants for utilization of solar energy in photosynthesis. It is also very important that in contrast with the pure carotenoid-TiO_2_ system, which allows carotenoid radical cations to be detected only at low temperatures (77 K), in the case of the carotenoid-AG complex, a significant increase in the stability of the radical cation was detected. Increasing the temperature up to room temperature does not lead to the disappearance of the spectrum. The lifetime of the canthaxanthin radical cation measured in [40] is approximately 10 days at room temperature. An increased stability of the carotenoid radical cation incorporated into a polymer host opens wide possibilities for the application of these complexes in the design of artificial light-harvesting, photoredox and catalytic devices. Additionally, the stabilization of carotenoid radical cations might be important for the regulation of the antioxidant/pro-oxidant activities of carotenoids. The stability of radical cations is a very important factor for the antioxidant activity of carotenoids. On the other hand, as mentioned above, the deprotonation of radical cations results in the formation of highly reactive neutral radicals. Yang, with co-authors, demonstrated that the generation of oxygen-irrelevant neutral radicals of carotenoids in the tumor microenvironment can offer novel opportunities to maximize the efficacy of chemodynamic therapy [84]. In this study, the authors demonstrated that proton-coupled electron transfer can promote the generation of neutral C-centered radicals of astaxanthin. The free radicals burst can significantly elevate free radical stress and induce cancer cell apoptosis.

### 4.3. Several Examples of Using Supramolecular Delivery Systems for Rare Carotenoids

In contrast to commonly used carotenoids, such as β-carotene, lutein, zeaxanthin, lycopene and astaxanthin, there are very few examples of using supramolecular delivery systems for rare carotenoids [43]. Esposito, with co-authors, developed lipid nanocarriers (ethosomes and organogels) for the cutaneous administration of crocin [85]. In vivo studies based on tape stripping and skin reflectance spectrophotometry have shown a more rapid anti-inflammatory effect and a rapid penetration of crocin from lipid nanocarriers, probably due to a strong interaction between phospholipids in the carrier and the lipids present in the stratum corneum. Since crocin has been shown to have the potential of inhibiting tumor genesis in a variety of cancers, there were attempts to use several drug delivery systems to improve its antitumor efficacy. For example, magnetite nanoparticles (MNPs) coated with natural polymers such as dextran have emerged as a promising technology for the targeted delivery of crocin to its site of action [86,87]. Coated MNPs can reduce the uptake by phagocytic cells, prolong the circulation time and passively accumulate in the tumor, leading to improvements in the therapeutic effect of crocin.

## 5. Conclusions

This paper predicts the behavior of some rare carotenoids that contain more oxygen groups than the conventional carotenoids or that are placed in positions other than on the rings. By careful design or self-assembly, these rare carotenoids can form multiple H-bonds and coordination bonds in host molecules. The stability, oxidation potentials and antioxidant activity of the carotenoids can be improved, and the photo-oxidation efficiency of the carotenoids can also be controlled. The photostability of the carotenoids can be increased if the carotenoids are embedded in a nonpolar environment when no bonds are formed.

Due to their lipophilicity, carotenoids also easily form “host-guest”-type supramolecular inclusion complexes with water-soluble compounds that have a suitable hydrophilic surface and hydrophobic interior. The incorporation of carotenoids results in noncovalent binding between the “guest” carotenoid and the “host” macromolecule and can significantly change the physical and chemical properties of carotenoids. Studies of the complexation of carotenoids with glycyrrhizin and arabinogalactan show that complexation affects the aggregation ability of some xanthophylls, their photostability and antioxidant activity. We have presented here our expertise on several conventional carotenoids studied by EPR and DFT and discussed the possible chemistry for some other several rare carotenoids, but endless possibilities remain for the approximately 1000 carotenoids left to be studied.

## Figures and Tables

**Figure 1 ijms-24-09885-f001:**
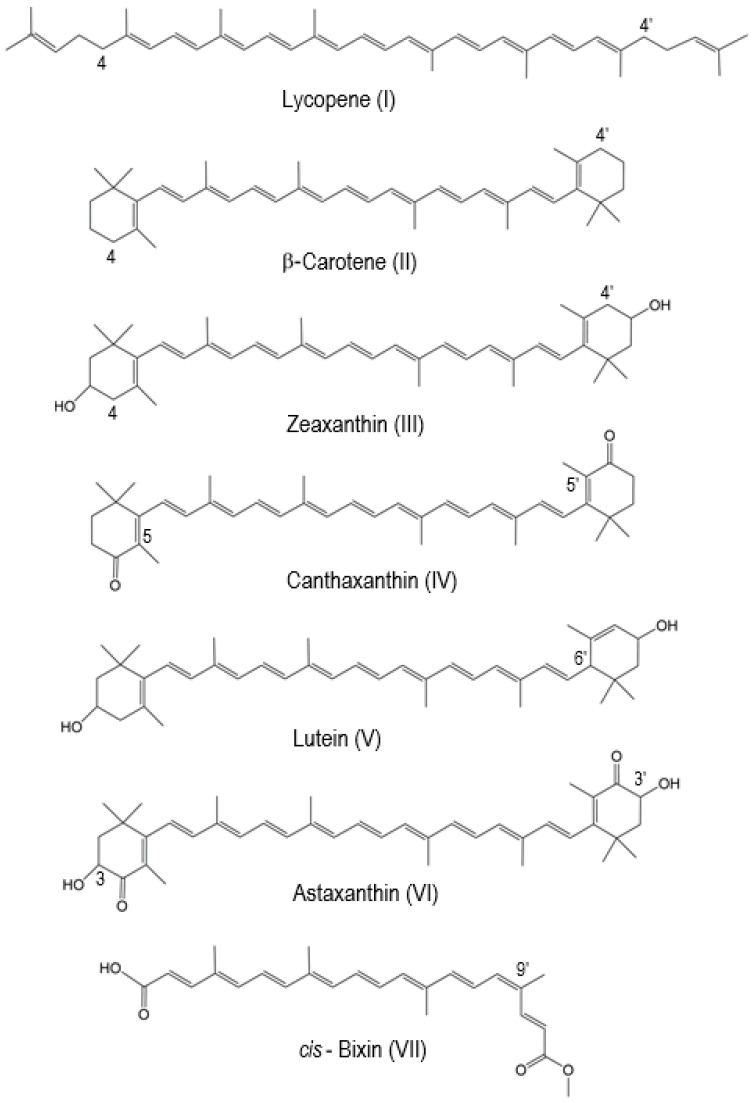
The structures of selected natural carotenoids.

**Figure 2 ijms-24-09885-f002:**
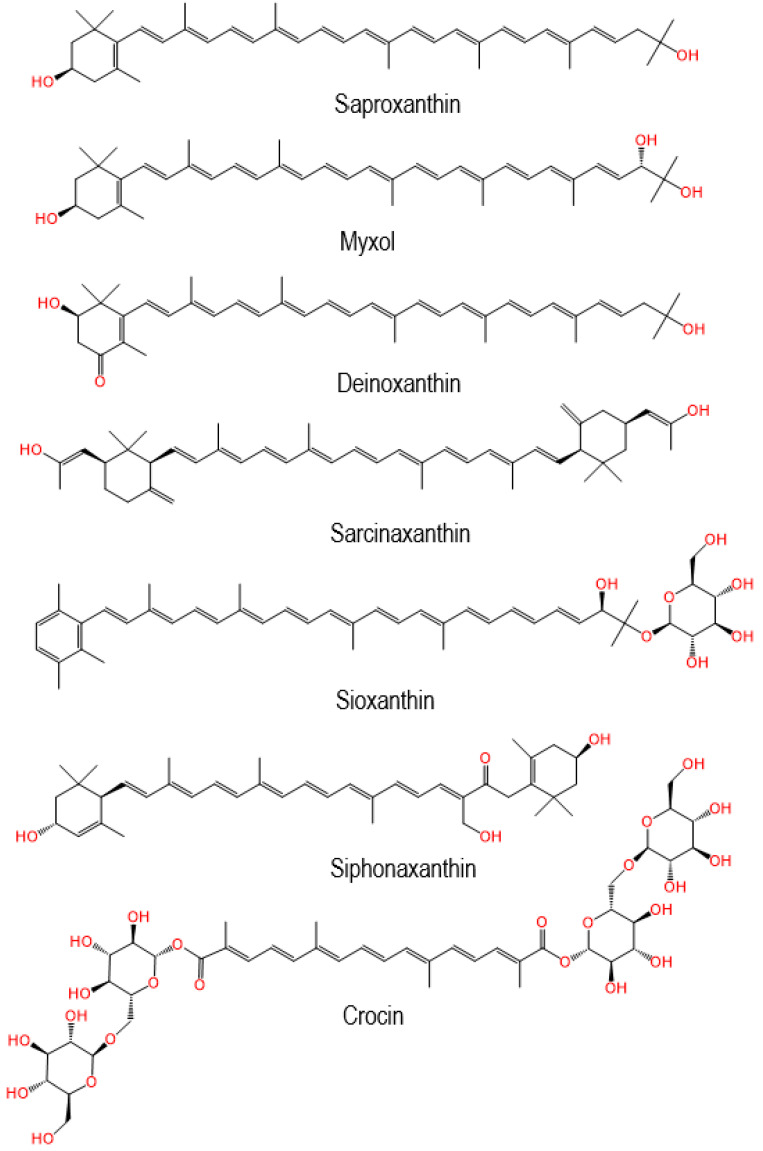
The structures of selected rare carotenoids.

**Figure 3 ijms-24-09885-f003:**
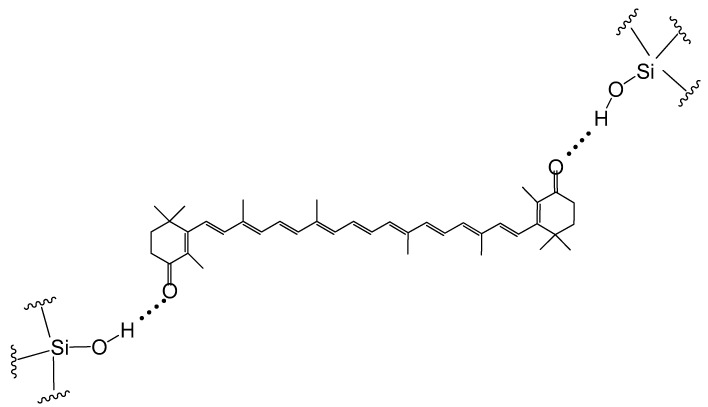
Formation of H-bonds of canthaxanthin with silanol groups on the surface of MCM-41.

**Figure 6 ijms-24-09885-f006:**
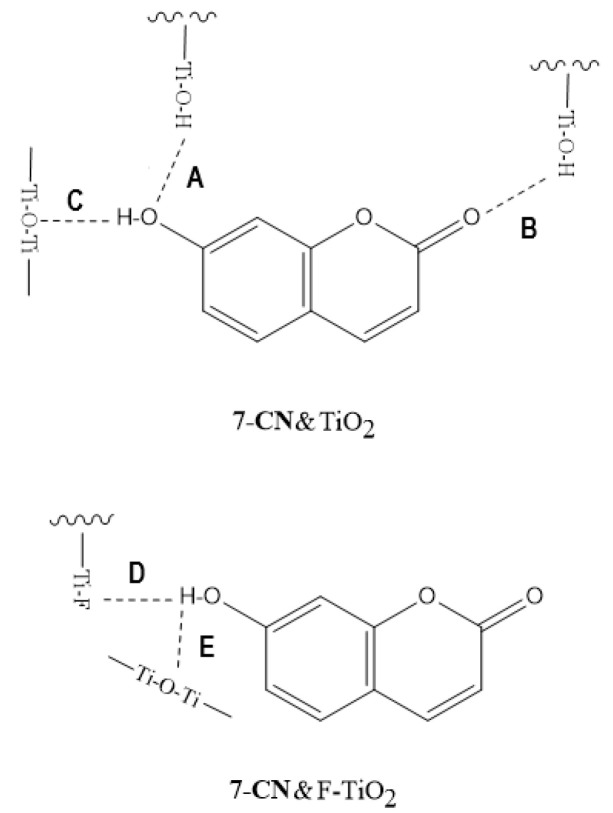
The formation of H-bonds of 7-CN on TiO_2_ and F-TiO_2_. 7-CN acts as both H-bond donor and acceptor on TiO_2_, but acts as only H-bond donor on F-TiO_2_. Adapted from Ref. [36].

**Figure 7 ijms-24-09885-f007:**
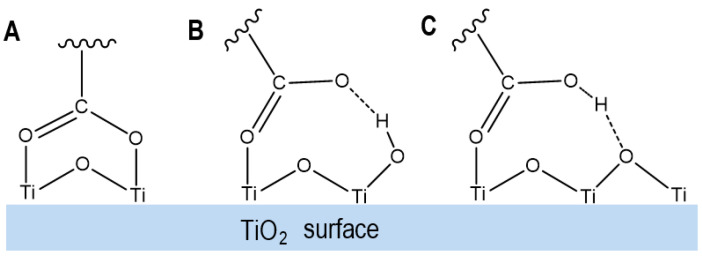
Three anchoring modes of a carboxylic acid group on the surface of TiO_2_: (**A**) the carboxylic acid group loses a proton and forms two coordination bonds with two surface Ti atoms of TiO_2_; (**B**) the carboxylic acid group loses a proton and forms one coordination bond with one surface Ti atom of TiO_2_ and one hydrogen bond (H-bond) with a Ti-OH group on the surface of TiO_2_; (**C**) the carboxylic acid group does not lose a proton and forms one coordination bond with one surface Ti atom of TiO_2_ and one H-bond due to the interaction between the OH group of the carboxylic acid group and the bridging oxygen atom on the surface of TiO_2_. Adapted from Ref. [37].

**Figure 8 ijms-24-09885-f008:**
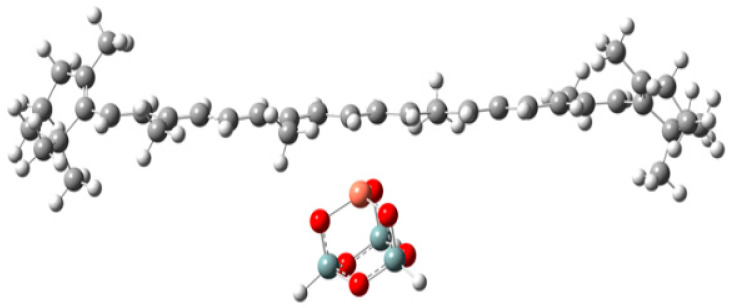
The optimized structures of Cu^2+^-β-carotene complex by B3LYP/6-31G(d). H: light gray, O: red, Cu: orange and C: dark gray. Si: blue gray. Adapted from Ref. [39].

**Figure 9 ijms-24-09885-f009:**
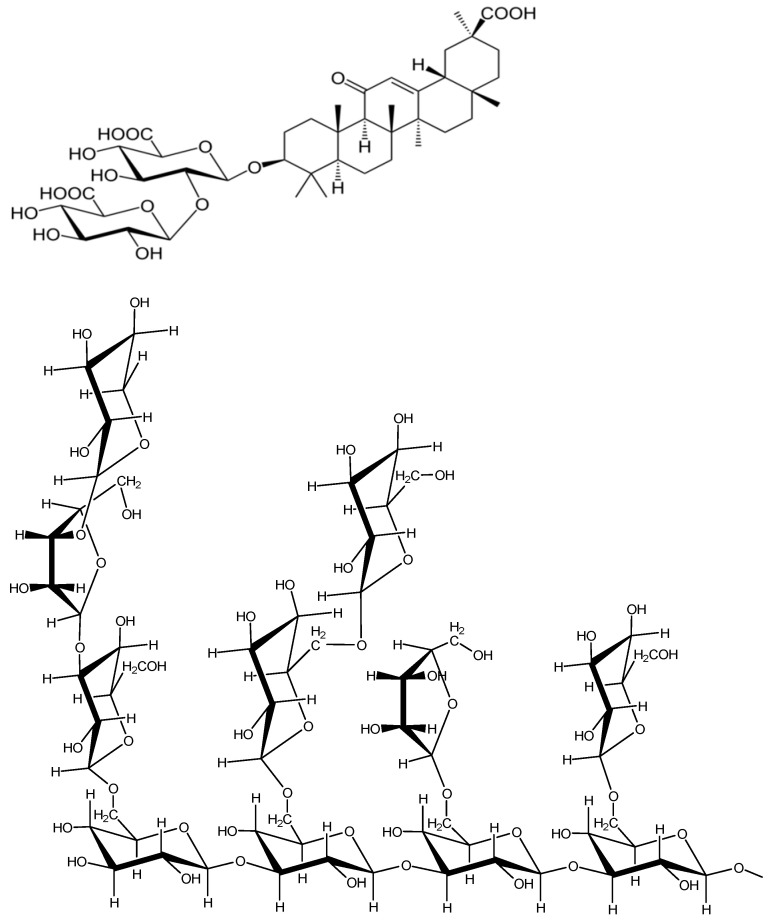
The structures of glycyrrhizin molecule (GA) (**top** image), and fragment of a branched polysaccharide arabinogalactan (AG) (**bottom** image).

**Table 2 ijms-24-09885-t002:** Biological roles of rare carotenoids.

Rare Carotenoid	Biological Role	Reference
Siphonaxanthin	anticancer activity in the treatment of leukemia anti-angiogenic, antioxidant and anti-inflammatory activities	[53] [54,55]
Saproxanthin	potent antioxidant	[56,57]
Myxol	potent antioxidant strengthening of biological membranes, reducing permeability to oxygen	[56,57] [58]
Sioxanthin	role in oxidative stress prevention/membrane fluidity	[56,62]
Deinoxanthin	strong ROS-scavenging activity potent inducer of apoptosis in human cancer cells, possible pro-oxidant activity	[59] [60]
Crocin	antioxidant, learning and memory enhancer, activity against brain neurodegenerative and Alzheimer disorders potent inducer of apoptosis and anti-angiogenesis effect	[63] [64]
Sarcinaxantin	high antioxidant activity for singlet oxygen (^1^O_2_)	[65]

## Data Availability

Not applicable.

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
