# Peer review of "The Endless World of Carotenoids—Structural, Chemical and Biological Aspects of Some Rare Carotenoids"

_ijms, 2023, doi:10.3390/ijms24129885_

Round 1

Reviewer 1 Report

This manuscript covered relevant and interesting information regarding the physicochemical properties of selected rare carotenoids by EPR and DFT studies. The manuscript has merit to be published after some important considerations:

Title: I strongly suggest revision for a more specific approach as “the endless world of carotenoids” do not fitted to the content.

Abstract – Please, provide the names of the selected rare carotenoids that will be approached in this review and explain the reason they are being considered rare.

References in the text – why were the references provided in Roman numbers?

Introduction – I consider that one of the most relevant biological property of selected carotenoids must be included during the introduction section: provitamin A activity (explain the structural requirements)

Lines 31-37 – provide proper strong references to these important statements.

Line 42 – As you indicated the meaning of EPR, please provide the meaning of DFT (Density functional theory) as well.

Lines 52-72 – Is it available only one reference to support all the information provided in these lines? Please, revise, and add supporting references to the subject.

Lines 126-129 – The authors must provide the complete explanation about the higher stability found for cis-bixin in comparison to trans-bixin. Please, indicate the true reference that stated the information. Reference IX is only mentioning another reference.

Lines 151-152 – Better explanation about the chose should be provided as saproxanthin and sarcinaxanthin have two OH groups as the previous mentioned in the Figure 1.

Lines 285-290 – Provide reference to support the information regarding deinoxanthin interactions on the surface of a semincondutor.

Section 3 – Is there any steric hindrance to mention for these carotenoids concerning the reported interactions? If so, please add in the text.

Section 4. Lines 298-300 – The authors must be clearer regarding the factor that affect carotenoids bioavailability. It is not as simple as it was mentioned.

Please, see: J.J.M. Castenmiller&C.E. West. Bioavailability and bioconversion of carotenoids. Annu. Rev. Nutr. 1998, 18:19–38.

Lines 322-324 – Provide references to support the following information: “… strong antioxidant activity of carotenoids reduces the risk of many types of diseases caused by free radicals, such as cardiovascular disease, cancer, and other age-related diseases”.

Section 4.1 – A Table summarizing the known biological activities of these rare carotenoids would add valuable contribution to the review.

Author Response

Changes in the manuscript appear in red in the word file. Conversion of word file into mdpi caused conversion of references into Roman numerals. At this point, we needed to send the comments in time, so please consider this word file.  We will send the mdpi file when complete.

Reviewer 2 Report

Dear Editor

The manuscript is well-written, interesting  and includes a vast overview of the subject that will contribute to the literature.

The only minor aspect is the references which I am not sure whether it is a mistake of the journal request (there are two numbering systems next to  each ref.)

Author Response

Changes in the manuscript appear in red in the word file. Conversion of word file into mdpi caused conversion of references into Roman numerals. At this point, we needed to send the comments in time, so please consider this word file.  We will send the mdpi file when complete. We apologize for any inconvenience this may have caused.

Reviewer 3 Report

The paper surveys the EPR and DFT studies relevant to understanding the carotenoid chemistry. In this sense, I did not understand why the paper was announced as a research article and not a review? In general, in the current state, this manuscript represents a limited review basically leaning on the papers of Dr. Kispert and colleagues. Although the manuscript is well written and covers most of the Dr. Kispert’s works, it hardly can be considered to be sufficient for publishing in the Q1 journal like IJMS. The topic is original and, apparently, would be interesting for wide audience of IJMS. In this way, it deserves rather more thorough consideration than one can see in the presented manuscript.  I suggest a significant extension of this paper with the critical consideration of the works of other researches involved in the EPR investigations and DFT calculations of carotenoids, like those of Donatella Carbonera et al. (https://doi.org/10.1021/ja9712074) or Elizabeth Hernandez-Marin et al. (https://doi.org/10.1021/jp401647n), etc. As to specific improvements, please, fix the issue with double numbering of references in the end of the manuscript and made the Arabic enumeration for the references in the text.  

Author Response

(The authors gave the same response as above.)

Round 2

Reviewer 1 Report

The authors accepted/considered all of the suggestions/comments and the now the manuscript is ready to be published.

Reviewer 3 Report

The authors answered to all my questions and made appropriate alternations. I recommend to accept this review for publication in IJMS.